# Reasons for encounter in primary care among French patients: Gender differences in presentation rates more pronounced among the patients of male practitioners

Eugenia Alcalde [1], Laurent Rigal[1,2], Henri Panjo[1], Laurent Letrilliart[3,4], Colinne Patrice [1,2]*

1 Université Paris-Saclay, UVSQ, Inserm, CESP U1018, Villejuif, France, 2 Département de Médecine Générale, Université Paris-Saclay, Univ. Paris-Sud, Le Kremlin-Bicêtre, France, 3 Université Claude Bernard Lyon 1, Research on Healthcare Performance (RESHAPE, U1290), Lyon, France, 4 Université Claude Bernard Lyon I, Faculté de Médecine Lyon Est, Lyon, France

* cocopatri@hotmail.com

## Abstract

### Background

Evidence has shown that the gender of general practitioners (GPs) and of patients plays a role in primary care, both in communication and practice style. However, less is known about how both genders interact in the reasons for encounter (RfEs). Studying RfEs by both patient and GP gender may help elucidate what symptoms or complaints lead women and men to consult a GP and specifically, if patients have preferences regarding the gender of their GP for specific problems. Given that men tend to consult less frequently and are often diagnosed at later stages of disease, it is essential to disentangle whether GP gender may act as a barrier for men in disclosing certain health concerns.

### Objective

To examine the influence of patients' and GPs' gender on the presentation of clinical RfEs.

### Methods

This is an ancillary study from the French ECOGEN survey, an observational cross-sectional multicentre study conducted between November 2011 and April 2012. 54 GP interns working alongside 128 GP supervisors collected RfEs during consultations. Using mixed models, we first assessed the association between patient gender and each RfEs chapter (i.e., 13 dependent variables). We then stratified models according to GP gender. Finally, we compared the four GP–patient gender dyads

**Data availability statement:** The data used in this study contain sensitive health information and are subject to legal restrictions under French data protection regulations. As required by the French Data Protection Authority (CNIL), these data cannot be publicly shared. Qualified researchers may request access to the minimal dataset from the RESearch on HealthcAre Performance (RESHAPE) research unit of the University Lyon 1 (reshape@univ-lyon1.fr). The institution guarantees persistent data storage and is responsible for maintaining availability for qualified researchers who meet the legal and ethical requirements.

**Funding:** The ECOGEN study was supported by the French National College of teachers in general practice via a grant from Pfizer laboratories (grant number not applicable), received by Laurent Letrillart. Eugenia Alcalde was funded by the Gender and Health Inequalities (GENDHI) project (grant agreement No. [ERC-2019-SyG n° 856478]). This funding had no role in study design, data collection and analysis, decision to publish, or preparation of the manuscript.

**Competing interests:** The authors have declared that no competing interests exist.

(male GP–male patient, male GP–female patient, female GP–male patient, and female GP–female patient) in pairwise analyses.

## Results

Overall, 20 613 patients were included, of whom 58% were women. Of the 128 participating GPs, 85 were men. Women had higher presentation rates across all chapters. The most common RfEs were those related to the respiratory system. Modest differences in presentation rates were observed between men and women when visiting a man GP, with men being less likely to present RfEs from the General and Unspecified, Digestive, Eyes, Ears, Neurological, and Respiratory chapters.

## Conclusions

Our study adds evidence on how gender plays a role in presenting health problems in primary care. Specifically, our findings suggest that men may be reluctant to present certain RfEs to men GPs. Acknowledging these gender effects in primary care may help strengthen the partnership between GPs and patients.

## Introduction

It has been largely documented that women have more contact with the health system over the life course than men. In many countries, epidemiological and routinely collected data show that women consult their general practitioner (GP) more frequently than men [1,2]. This can be partially explained by gynecological reasons and family planning visits. Likewise, it can be related to differences in disease prevalence rates between women and men. Women experience more non-fatal chronic diseases, such as arthritis and depression [3], which are common health problems managed by GPs. In contrast, men experience more fatal diseases, such as heart disease and stroke [3].

Research studying this gender gap in primary care use is mixed. Some studies have found that this disparity still remains even after controlling for health status [1]. However, a study conducted in the UK using routinely collected data showed that, among patients with common underlying comorbidities (e.g., cardiovascular disease and depression) and also among the oldest, the gap was reduced and modest [2]. A literature review indicated that, although gaps and inconsistencies remain across studies comparing men's and women's help-seeking care, traditional masculinity — shaped by early gender socialization— appears to contribute to men's reluctance to seek help when they experience illness [4]. In that sense, the gender gap in primary care use cannot be explained only by differences in prevalence, as gender norms and gender-related health beliefs also seem to contribute to men's lower and delayed use [5,6].

One interesting way to analyse this gender gap is to investigate reasons for encounter (RfEs), particularly symptoms and complaints, as they reflect what

prompts men and women to seek care and what they decide to disclose to their GPs. In other words, RfEs provide insight into how individuals evaluate and prioritize health concerns in their decision-making process to seek help. Until now, only limited research has been conducted on RfEs according to patients' gender. An Israeli study showed that men were more likely to visit the GP for known and chronic problems, whereas women were more likely to consult for new medical problems [7]. The same study showed that men were more likely to consult a male GP for new problems and anxiety, but were more likely to see a female GP for follow-up care for chronic illness [7].

With the feminization of the healthcare workforce, the influence of GPs' gender has been studied across a wide range of aspects of medical care, and it is now widely accepted that it affects the process of care. In brief, female GPs have been described as adopting a 'patient-centred' communication style, while male GPs tend to adopt a 'technical' one [8]. Female GPs conduct more preventive consultations [9] and psychosocial counselling sessions, and order more analyses and referrals than their male counterparts [10]. It has been documented that patients are generally more satisfied with interactions involving female GPs [8]. Moreover, GP-patient gender concordance has been described as a determinant for prevention procedures in general practice [11] and in cancer screening [12]. Studies investigating RfEs patterns by GP's gender have reached similar conclusions, with a few exceptions. Female GPs see more patients with RfEs relating to the general and unspecified, endocrine, urinary, reproductive and female genital systems [13–15], as well as psychological and social issues [13,14]. Conversely, female GPs see fewer patients with RfEs relating to the musculoskeletal, respiratory, male genital, skin, eye, and ear systems [13–15]. Importantly, these findings suggest that patients' preferences for the gender of their GP are not limited to intimate or genital-related consultations, but extend to a wide range of health problems.

In France, GPs act as the gatekeepers of the healthcare system, meaning that people are encouraged to visit a GP first, who will then refer them to specialized care if needed. People are also required to register with one GP (or other physician specialty), a measure intended to ensure the coordination and continuity throughout the care pathway. Unlike in other countries with gatekeeper systems, the French system adopts a liberal approach: people can choose to consult a different physician than the registered one, although they will be less reimbursed by the social security system [16]. This implies that, to some extent, people can choose any GP for any given consultation. Even though many factors can play in this choice, starting from availability, we advance that, when possible, people may choose a GP for his or her gender for a specific problem.

Previous studies investigating the effect of patients' and GPs' gender on RfEs usually present results based on one or the other, but rarely together. As both seem to influence the care process, the challenge lies in distinguishing how they operate within a wide range of RfEs presented in primary care. Given that men consult less frequently and are often diagnosed at later stages of disease, it is important to disentangle whether GP gender may act as a barrier for men in disclosing certain health problems. Using observational data collected in French primary care practices, this study aimed to examine the combined effect of gender on the presentation of clinical RfEs. We hypothesise that GPs' gender can affect presentation rates for men and women patients, resulting in differences in the presentation of certain health complaints according to the gender of the GP.

## Methods

### Study design

We used data from the French ECOGEN survey, an observational cross-sectional nationwide multicentre study conducted from 28 November 2011–30 April 2012 across 128 GP's offices. This ancillary analysis was conducted over the period from May to July 2020. The ECOGEN survey was designed to describe the clinical activity of French GP, including the health problems managed, the associated RfEs and processes of care [17]. The participating doctors supervised GP interns and were attached to one of the 27 partner French medical schools. Each intern was placed in two or three different practices and no practice had more than one intern.

The survey included all home and office visits of the participating GPs, taking place on determined half-days shifts per week distributed across the study period, for a total of 20 consultation-days per GP. Patients were given oral and visual information about the study, including the presence of the intern. Verbal consent was obtained from patients or the parents of minor participants. Patients could refuse the intern's presence while still participating in the study, or refuse both. Visits for which patients refused participation were excluded. No incentives were given to patients, interns, or GPs for participating in this study.

No directly identifying data was collected in the database, and authors had no access to any identifying data. A declaration was filed with the former Advisory Committee on the Processing of Information in Health Research (CCTIRS, active at the time ECOGEN was designed) and then with the National Commission for Information Technology and Civil Liberties (CNIL) – both advisory boards approved the research methodology. Further, the ECOGEN survey was approved by Ethics Committee Sud-Est IV (No.L11-149), which explicitly approved the oral form of consent obtained from patients given the observational and minimal-risk nature of the study. In accordance with the Ethics Committee's requirements, no written documentation or witness signature of the oral consent was requested or required.

## Data collection

Data were collected by 54 interns, observing their supervisors' general practice. They were trained for this data collection, including in the use of the International Classification of Primary Care (ICPC-2) [18]. Data was collected on paper forms at the end of each encounter and then was entered daily in a secure central database via a dedicated website.

Data regarding the characteristics of the included GPs included age, sex, fees (that is, fixed fees or authorisation for fees beyond the amount reimbursed by the national health insurance fund), mode of practice (solo, group, private multi-professional, or public health centre), practice location, number of consultations per year. The data concerning the consultations included the patient's age, sex, socio-professional category, health insurance status (specifically, exemption from fees due to low income or serious chronic disease), visit site (office or home), health problems managed and the associated RfEs and processes of care, and consultation length. The RfEs, the health problems managed and the processes of care performed were coded according to the ICPC-2 classification, with the support of a coding engine system [19]. The ICPC-2 is organized in 17 chapters: 15 based on body systems for somatic health problems (General and Unspecified; Blood; Digestive; Eye; Ear; Cardiovascular; Musculoskeletal; Neurological; Respiratory; Skin; Metabolic/endocrine and nutrition; Urological, Pregnancy, family planning; Female genital; Male genital), one for psychological problems and one for social problems. It also includes the following 7 components: Symptoms, complaints; Diagnostics, screening prevention; Treatments, procedures, medication; Test results; Administrative; Referrals and other reasons for encounter; Diagnoses, diseases.

## Statistical analyses

The 13 dependent variables analysed were the presence in the consultation of at least one RfEs belonging to each of the ICPC-2 chapters (chapters Pregnancy, family planning, Female and Male genital have not been considered as they relate to only one sex, and Blood being excluded due to insufficient numbers to carry out the analysis). Moreover, only the reasons belonging to symptoms and complaints have been retained to study symptoms that were new or not yet stabilised.

For each RfEs chapter, we performed multivariable analyses using logistic mixed-effect models with random intercepts for GP effect and including two levels: the GP, and consultation [20]. In order to adjust for potential confounders, all models were adjusted for patients' characteristics significantly different between patients' gender and, GPs' characteristics significantly different between GPs' sex in univariable analysis.

The associations between patient gender and each RfEs chapters were estimated first in all patients regardless of GP gender (overall multivariate analysis) and second by stratifying the sample according to the gender of each patient's GP (stratified analysis). We then compared the four types of GP–patient gender dyads (male GP-male patient, male

GP-female patient, female GP-male patient, and female GP-female patient) in a pairwise analysis using hierarchical logistic regression models.

The statistical analyses were performed with SAS software v.9.4.

## Results

A total of 128 GP supervisors participated in this study, 85 of whom were men (Table 1). Male GPs had a significantly higher mean annual number of consultations than their female GP counterparts.

A total of 20 613 patients (Table 2) were included after exclusion of 168 who refused the presence of the intern in the consultation room. Included patients were mostly women. Male patients were younger than female patients. Most of the patients did not have an exemption from medical fees for a serious chronic disease. Slightly over half (51.6%) of the 20 613 GP/patient dyads were gender concordant: 29.6% were male concordant while 22.0% were female concordant.

On average, 2,6 RfEs were recorded per consultation. Women had higher presentation rates across most RfEs chapters (Table 3). The most common RfEs chapter encoded were related to the Respiratory system, accounting 1 794 for men and 2 493 for women patients, followed by Musculoskeletal and General/Unspecified RfEs chapters. In the regression analysis

**Table1. Participating GPs' characteristics by gender (N = 128).**

| | Total N = 128 n (%) or m (SD) | Male GP N = 85 n (%) or m (SD) | Female GP N = 43 n (%) or m (SD) | p |
|---|---|---|---|---|
| **Gender** Male | 85 (66.4) | | | |
| Female | 43 (33.6) | | | |
| **GP age [years]** | | | | 0.09 |
| 32–49 | 35 (27.3) | 18 (21.2) | 17 (39.5) | |
| 50–54 | 33 (25.8) | 21 (24.7) | 12 (27.9) | |
| 55–59 | 38 (29.7) | 29 (34.1) | 9 (20.9) | |
| ≥ 60 | 22 (17.2) | 17 (13.3) | 5 (3.9) | |
| **Annual no of consultations** | | | | 0.02 |
| 0–4999 | 74 (57.8) | 43 (50.6) | 31 (72.1) | |
| 5000–10500 | 54 (42.2) | 42 (49.4) | 12 (27.9) | |
| **Mean annual no of consultations** | | | | 0.0004 |
| | 5139 (1762) | 5525 (1725) | 4376 (1595) | |
| **Practice location** | | | | 0.43 |
| Rural areas | 29 (22.7) | 22 (25.9) | 7 (16.3) | |
| Semi rural | 33 (25.8) | 22 (25.9) | 11 (25.6) | |
| Urban areas | 66 (51.6) | 41 (48.2) | 25 (58.1) | |
| **Mode of practice** | | | | 0.46 |
| Solo | 27 (21.1) | 21 (24.7) | 6 (14.0) | |
| Group | 79 (61.7) | 49 (57.6) | 30 (69.8) | |
| Private multiprofessional | 20 (15.6) | 14 (16.5) | 6 (14.0) | |
| Public health centre | 2 (1.6) | 1 (1.2) | 1 (2.3) | |
| **Fees** | | | | 0.66 |
| Set by the health authorities | 118 (92.2) | 79 (92.9) | 39 (90.7) | |
| Set by the GP | 10 (7.8) | 6 (7.1) | 4 (9.3) | |

GP, general practitioner.

**Table 2. Patients' characteristics by gender (N = 20 613).**

| Patient characteristics | Men (n = 8618) No (%)/Mean (Sd) | Women (n = 11995) No (%)/ Mean (Sd) | p |
|---|---|---|---|
| **Age (years)** | | | **<.0001** |
| 0-14 | 1658 (19.2) | 1604 (13.4) | |
| 15-44 | 2246 (26.1) | 3697 (30.8) | |
| 45-74 | 3490 (40.5) | 4635 (38.6) | |
| ≥75 | 1224 (14.2) | 2059 (17.2) | |
| **Mean age (years)** | 44.9 (26.2) | 47.4 (25.3) | **<.0001** |
| **Socio-professional category** | | | **<.0001** |
| Farmer/craftsman/ shopkeeper/ business owner/ Executive, intellectual profession | 1356 (15.7) | 1554 (13.0) | |
| Office worker/ Manual worker | 1875 (21.8) | 2912 (24.3) | |
| Retired | 2754 (32.0) | 4012 (33.4) | |
| Unemployed | 2633 (30.5) | 3517 (29.3) | |
| **Exemption for medical fees** | | | **<. 0001** |
| At least one | 2877 (33.4) | 3370 (28.1) | |
| None | 5741 (66.6) | 8625 (71.9) | |
| **Exemption from medical fees for a serious chronic disease** | | | **<.0001** |
| Yes | 2226 (25.8) | 2534 (21.1) | |
| No | 6392 (74.2) | 9461 (78.9) | |
| **Exemption from medical fees for low income** | | | 0.08 |
| Yes | 358 (4.2) | 554 (4.6) | |
| No | 8260 (95.8) | 11441 (95.4) | |
| **Consultation place** | | | **<.0001** |
| GP's office | 8213 (95.3) | 11131 (92.8) | |
| Patient's home | 405 (4.7) | 864 (7.2) | |
| **GPs gender** | | | **<.0001** |
| Men | 6104 (70.8) | 7446 (62.1) | |
| Women | 2514 (29.2) | 4549 (37.9) | |
| **Already known patient** | | | **0.006** |
| No | 526 (6.1) | 614 (5.1) | |
| Yes | 8092 (93.9) | 11381 (94.9) | |
| **Consultation length (minutes)** 1-10 | 2227 (25.8) | 2801 (23.4) | **<.0001** |
| 11-15 | 2634 (30.6) | 3468 (28.9) | |
| 16-20 | 1787 (20.7) | 2547 (21.2) | |
| 21-30 | 1470 (17.1) | 2331 (19.4) | |
| >30 | 500 (5.8) | 848 (7.1) | |
| **Mean consultation length (minutes)** | 16.2 (18.6) | 17.5 (12.7) | 0.31 |
| **Mean number of RfEs** | 2.55 (1.6) | 2.73 (1.7) | **<.0001** |

by patients' gender, men were less likely to present general and unspecified reasons; problems related to the following systems: digestive, eyes, ears, neurological, psychiatric, respiratory and urological; and social issues.

Table 4 displays our stratified models by GPs gender. In both genders, Respiratory RfEs were the most encoded, followed by Musculoskeletal and General/Unspecified RfEs. Differences in presentation rates were more commonly observed between men and women visiting a male GP. Compared to female patients, male patients consulting a male GP

**Table 3. Distribution of reasons for encounter by patient's gender. Overall multivariate analysis.**

| Reasons for encounter (Symptoms/complaints) | Men N=8618 n (%) | Women N=11995 n (%) | ORᵃ (Ref: women) | P |
|---|---|---|---|---|
| General and Unspecified | 1109 (12.9) | 1652 (13.8) | 0.87 [0.80 - 0.95] | **0.002** |
| Digestive | 992 (11.5) | 1526 (12.7) | 0.90 [0.83 - 0.98] | **0.02** |
| Eye | 127 (1.5) | 215 (1.8) | 0.78 [0.63 - 0.98] | **0.03** |
| Ear | 219 (2.5) | 362 (3.0) | 0.76 [0.64 - 0.91] | **0.002** |
| Cardiovascular | 128 (1.5) | 200 (1.7) | 0.91 [0.72 - 1.14] | 0.40 |
| Musculoskeletal | 1454 (16.9) | 2179 (18.2) | 0.98 [0.91 - 1.06] | 0.57 |
| Neurological | 423 (4.9) | 744 (6.2) | 0.83 [0.73 - 0.94] | **0.004** |
| Psychological | 438 (5.1) | 710 (5.9) | 0.93 [0.82–1.06] | 0.26 |
| Respiratory | 1794 (20.8) | 2493 (20.8) | 0.92 [0.86 - 0.99] | **0.02** |
| Skin | 570 (6.6) | 798 (6.7) | 1.00 [0.90 - 1.13] | 0.93 |
| Endocrine/Metabolic and, nutritional | 102 (1.2) | 179 (1.5) | 0.83 [0.65 - 1.07] | 0.15 |
| Urological | 160 (1.9) | 240 (2.0) | 0.96 [0.78–1.18] | 0.71 |
| Social | 207 (2.4) | 396 (3.3) | 0.79 [0.66 - 0.95] | **0.01** |

ᵃAdjusted odd ratios for patient variables (Age, Socio-professional category, Exemption from medical fees for a serious chronic disease, Exemption from medical fees for low income and Consultation length (minutes), practice location and Established patient) and for GP variables (Age, sex, Mean annual number of consultations), followed by their 95% confidence intervals.

were less likely to present reasons corresponding to General and Unspecified, Digestive, Eyes, Ears, Neurological and Respiratory chapters. Among female GPs, differences between women and men patients were attenuated, except for Ear and Social chapters which were less presented by men.

Results from the pairwise analyses are available in S1 Table. These analyses showed that, globally, RfEs corresponding to General/Unspecified, Earing, Neurological and Social issues had differences in presentation rates depending on the gender dyad composition. Consistently with the stratified analyses, the male GP/male patient dyad was less likely to endorse the aforementioned chapters, compared to the other combinations.

## Discussion

### Summary

In this study, we explored how GP and patient gender are associated with RfEs, using observational data collected in the consultation room. Women had higher presentation rates across most RfEs chapters, regardless of GPs' gender. Although modest, the presentation rates of men and women visiting male GPs differed for almost half of the RfEs chapters. Men seemed less likely than women to present with certain RfEs chapters when the GP was male.

### Comparison with the literature

The higher presentation rates among women, regardless of the gender of the GP, are not surprising because they are consistent with epidemiological data. From a gender perspective, social roles and gender norms impact the health-seeking behaviour of both women and men. Qualitative research has shown that ideals of masculinity can affect men's experience of illness in many ways, including their reluctance to seek care and their discourse when visiting a doctor [21–23]. Literature based on masculinities' health supports the idea that men are socialised to be stoic and strong when experiencing pain and illness [24]. In a gender-normative context, consulting a GP may be perceived as a sign of weakness, especially if the patient does not consider the health problem to be 'serious' [21,25]. This patient gender effect could explain, at least partially, why men consult less.

**Table 4. Distribution of reasons for encounter by patients' gender, stratified on GPs' gender. Stratified analyses.**

| Reasons for encounter (Symptoms/complaints) | Male GP | | | | Female GP | | | |
|---|---|---|---|---|---|---|---|---|
| | Men N = 6104 n (%) | Women N = 7446 n (%) | OR [a] (Ref: women) | p | Men N = 2514 n (%) | Women N = 4549 n (%) | OR [a] (Ref: women) | p |
| General and Unspecified | 735 (12.0) | 1014 (13.6) | 0.82 [0.74 - 0.92] | **0.0003** | 374 (14.9) | 638 (14.0) | 0.98 [0.84 - 1.13] | 0.73 |
| Digestive | 679 (11.1) | 916 (12.3) | 0.89 [0.80–0.99] | **0.03** | 313 (12.5) | 610 (13.4) | 0.94 [0.81 - 1.09] | 0.43 |
| Eye | 83 (1.4) | 136 (1.8) | 0.72 [0.55 - 0.96] | **0.02** | 44 (1.8) | 79 (1.7) | 0.91 [0.62 - 1.33] | 0.61 |
| Ear | 151 (2.5) | 216 (2.9) | 0.77 [0.62 - 0.96] | **0.02** | 68 (2.7) | 146 (3.2) | 0.74 [0.55–0.99] | **0.05** |
| Cardiovascular | 94 (1.5) | 146 (2.0) | 0.81 [0.62 - 1.06] | 0.13 | 34 (1.4) | 54 (1.2) | 1.28 [0.82 - 1.99] | 0.28 |
| Musculoskeletal | 1038 (17.0) | 1336 (17.9) | 0.98 [0.89 - 1.08] | 0.68 | 416 (16.6) | 843 (18.5) | 0.97 [0.85 - 1.11] | 0.64 |
| Neurological | 288 (4.7) | 469 (6.3) | 0.76 [0.65 - 0.88] | **0.0004** | 135 (5.4) | 275 (6.1) | 1.03 [0.83 - 1.28] | 0.78 |
| Psychological | 313 (5.1) | 427 (5.7) | 0.93 [0.79–1.08] | 0.33 | 125 (5.0) | 283 (6.2) | 0.93 [0.75 - 1.17] | 0.55 |
| Respiratory | 1235 (20.2) | 1549 (20.8) | 0.90 [0.82 - 0.98] | **0.02** | 559 (22.2) | 944 (20.8) | 0.95 [0.84 - 1.08] | 0.45 |
| Skin | 375 (6.1) | 466 (6.3) | 0.97 [0.84 - 1.12] | 0.66 | 195 (7.8) | 332 (7.3) | 1.08 [0.90 - 1.30] | 0.42 |
| Endocrine/Metabolic and, nutritional | 64 (1.1) | 106 (1.4) | 0.76 [0.56 - 1.05] | 0.10 | 38 (1.5) | 73 (1.6) | 0.95 [0.64- 1.43] | 0.82 |
| Urological | 114 (1.9) | 136 (1.8) | 1.04 [0.81 - 1.34] | 0.76 | 46 (1.8) | 104 (2.3) | 0.81 [0.57–1.16] | 0.25 |
| Social | 143 (2.3) | 209 (2.8) | 0.86 [0.69 - 1.08] | 0.20 | 64 (2.6) | 187 (4.1) | 0.70 [0.52 - 0.95] | **0.02** |

[a]Adjusted odd ratios for patient variables (Age, Socio-professional category, Exemption from medical fees for a serious chronic disease, Exemption from medical fees for low income and Consultation length (minutes), practice location and Established patient) and for GP variables (Age, Mean annual number of consultations), followed by their 95% confidence intervals

We found that respiratory issues were the most common RfEs, followed by musculoskeletal issues and general and unspecified issues. International comparative studies have also identified respiratory issues as being among the most common RfEs, regardless of patients' gender or age [26,27]. Moreover, our resulting distribution was similar to that found by Adar et al., with a few exceptions. In their study, the authors found that, while women presented more respiratory and digestive RfEs, men presented more musculoskeletal RfEs and diabetes [7]. We believe that international comparisons remain difficult to make, as many factors can influence RfEs presentation rates among patients. These factors range from local gender stereotypes and norms to coding discrepancies and the characteristics of the healthcare system itself.

## Interpretation of our findings

Although the differences in RfEs by patient gender were modest when stratified by GP sex, some of our findings warrant further discussion. The most interesting result of our study was that men seem to go to women doctors more often than to men doctors for some health problems. In line with the findings of Bensing et al., this result demonstrates that patients have preferences regarding their GPs' gender for specific health complaints [13], and this is also the case for problems that are not physically intimate. It is unclear whether this relates to different practice styles, gender norms, or both. The former has already been documented [8] and may indicate a patient-oriented attitude among women GPs, encouraging men patients to disclose health complaints more often to women GPs. The latter, as described in literature on masculinity, is interesting as it illustrates how masculine norms influence men's discourse when consulting a men GP. Men patients may perceive men GPs as less 'caring' than female GPs, who are stereotypically and historically associated with caregiving roles [28] and therefore often essentialized as natural caretakers [29]. We believe that both of these explanations contribute to this double gender effect and need to be addressed by increasing gender awareness in the medical training curricula and, more generally, in society.

From our stratified analysis, we observed that women presented more social complaints to women GPs. However, no differences were observed in this regard among patients visiting men GPs. Considering that this chapter includes

complaints related to violence and financial difficulties, we believe this finding indicates that women feel more comfortable disclosing these issues to women GPs. It is also interesting to relate these preferences to the practice style of women GPs (i.e., more patient-oriented), as previously documented by Harrison et al [15].

## Implications

If GPs' gender had been neutral, it would imply that men and women have the same presentation rates across both GP strata. However, since men appear to prefer women GPs for certain issues, the question arises as to the extent to which the nature of these RfEs is associated with gender norms. Bensing et al. suggested that GPs attract not only gender-concordant patients, but also specific health problems, regardless of the patient's gender, that could be associated with gender role stereotypes [13]. For example, in their study, men GPs were more likely to see RfEs corresponding to 'masculine' health problems, such as musculoskeletal issues, which the authors linked to sports-related injuries or work-related accidents [13]. In our study, the difference in presentation rates between patients of men and women GPs could be linked to the fact that men visiting a men GP could, consciously or unconsciously, avoid disclosing health complaints that are often stereotyped as feminine, such as migraines (coded in the neurological chapter) or coughing (coded in the respiratory chapter), because of concerns about 'manhood'. In this sense, GPs should pay attention to what men patients bring to, and do not bring to, the consultation room. Reassuring men and understanding their perspective may be especially important in men-only dyads.

## Strength and limitations

Our two-step approach (overall and stratified analyses) enabled us to identify patients' preferences regarding GPs' gender when consulting for new symptoms or complaints in primary care in a novel way. Our interpretation of our findings through the lens of gender, meets the growing demand for gender awareness in public health and medicine research fields. Gender is a major determinant for accessing health services and, in many situations, gender norms can be a barrier and a source of inequality that need to be addressed. Therefore, identifying the possible underlying mechanisms that operate in the presentation of a health complaint in consultation, such as the interaction between a male GP and his male patient or the stereotypes associated with some RfEs, can represent a leverage for filling the gap between men and women in primary care utilization.

This study has a number of limitations. The cross-sectional design does not allow any causal inference. Participating GPs were conveniently enrolled, making generalizability difficult to determine. However, the ECOGEN GP sample was representative of French GP in gender distribution, mean age, mean number of consultations per year, practice location and type of fees authorized [30]. Moreover, another French study showed that patients attending training practices are globally similar with the rest of the patients attending regular practices [31]. Regarding our analyses, for simplicity reasons, our models were adjusted for patients' age, but this did not allow us to assess the interaction between patients' age and gender when presenting a given RfEs. Indeed, it is possible that younger men or women have different attitudes towards GPs when presenting their health problems compared to older counterparts. We believe further studies should address this gap, especially considering how cohort effects regarding gender attitudes have changed in recent years. Also, other variables that could have explained gender differences in presentation rates were not measured, such as the number of times patients had previously visited a given GP or the RfEs for prior consultations. Similarly, the gender of the intern was not reported in the survey, so we could not assess whether it influenced the RfEs. Nevertheless, patients could refuse the presence of the intern, which would limit the risk of bias in reporting RfEs to their GPs. Finally, the analysed data were collected over 10 years prior to our study. Since then, many changes in society and in medical demographics have occurred, including a potential increase in gender awareness among patients and GPs, as well as the continued feminization of the GP population [32], which makes our results less generalizable to the current context. At the same time, the number of GPs in France declined in recent years until 2021 [33]. Although more recent data suggest a slight

increase since then [32], the reduced density of GPs per capita [34] may lead people to consult a GP based on availability. For these reasons, the gender of GPs may have less impact on RfEs than before, as people have fewer options. Still, using data from the early 2010s, when this density was higher than it is nowadays (155 GPs and 146 GPs per 100 000 persons in 2012 and 2025, respectively [35]), can be more reliable for exploring gender effects in the presentation of RfEs in primary care. In this regard, we argue that our findings, though not current, may be closer to true patient preferences, as access was easier back then.

## Conclusions

Women had higher presentation rates across most RfEs, regardless of the gender of the GP. We observed modest differences in presentation rates among patients who visited men GPs. Nevertheless, men patients were less likely to present with certain RfEs when the GP was a man. This finding suggests that men patients may have preferences regarding their GP's gender for certain symptoms or complaints. They may choose a women GP for certain issues, or avoid disclosing certain concerns to men GPs. Therefore, it is important to continue efforts to increase gender awareness among GPs and to consider how this awareness can affect the use of healthcare services. Future studies should consider adopting a longitudinal design to explore how GP-patient gender dyads evolve over time, particularly once a specific health complaint is resolved. This could shed light on whether the preferred GP gender remains consistent beyond the original RfEs.

## Supporting information

**S1 Table. Distribution of reasons for encounter by GP/patient gender dyad. Pairwise analyses.**
(DOCX)

## Author contributions

**Conceptualization:** Laurent Rigal, Laurent Letrilliart, Colinne Patrice.

**Data curation:** Laurent Letrilliart.

**Formal analysis:** Henri Panjo, Colinne Patrice.

**Investigation:** Laurent Letrilliart.

**Methodology:** Laurent Rigal, Laurent Letrilliart, Colinne Patrice.

**Project administration:** Laurent Letrilliart.

**Supervision:** Laurent Rigal.

**Writing – original draft:** Eugenia Alcalde.

**Writing – review & editing:** Eugenia Alcalde, Colinne Patrice.

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
