## [Decision Letter · Decision Letter 0]

8 Oct 2025

Dear Dr. Patrice,

Thank you for submitting your manuscript to PLOS ONE. After careful consideration, we feel that it has merit but does not fully meet PLOS ONE’s publication criteria as it currently stands. Therefore, we invite you to submit a revised version of the manuscript that addresses the points raised during the review process.

We look forward to receiving your revised manuscript.

Kind regards,

Academic Editor

PLOS ONE

Journal Requirements:

https://journals.plos.org/plosone/s/file?id=wjVg/PLOSOne_formatting_sample_main_body.pdf   and  and

3. In this instance it seems there may be acceptable restrictions in place that prevent the public sharing of your minimal data. However, in line with our goal of ensuring long-term data availability to all interested researchers, PLOS’ Data Policy states that authors cannot be the sole named individuals responsible for ensuring data access (http://journals.plos.org/plosone/s/data-availability#loc-acceptable-data-sharing-methods).

Reviewers' comments:

Reviewer's Responses to Questions

**Comments to the Author**

1. Is the manuscript technically sound, and do the data support the conclusions?

Reviewer #1: Partly

Reviewer #2: Partly

Reviewer #3: Yes

2. Has the statistical analysis been performed appropriately and rigorously?

Reviewer #1: Yes

Reviewer #2: Yes

Reviewer #3: Yes

3. Have the authors made all data underlying the findings in their manuscript fully available?

Reviewer #1: Yes

Reviewer #2: Yes

Reviewer #3: Yes

4. Is the manuscript presented in an intelligible fashion and written in standard English?

Reviewer #1: Yes

Reviewer #2: Yes

Reviewer #3: Yes

Reviewer #1: The study topic is good, but it is not completely clear and needs to be clarified, especially with regard to the study summary.

The study needs to detail the methodology, the summary, and even the results, and why gender affects medical issues.

Reviewer #2: The article provides a relevant analysis of gender differences in reasons for encounter in primary care, based on a large and representative dataset from France. The findings are valuable and the manuscript is well structured.

However, it is important for the authors to clarify that the results are not directly generalizable to the present context, since gender ideologies and the distribution of male and female physicians have changed in the last decade. In addition, the manuscript would benefit from a deeper consideration of the interaction between gender and age, for example discussing whether younger men and older men differ in consultation patterns, or explaining why this aspect was not explored.

Overall, the study is suitable for publication and represents a relevant contribution, but it would be strengthened by these additions to the discussion.

Reviewer #3: Abstract:

1. Line 27- You should show the (abbreviation) for general practitioner (GP). Line 31—You use GP, but it is not defined earlier.

2. Line 34 – Capitalize “To.”

Introduction

1. Recommend that authors provide background on the French healthcare system. For example, do patients have the flexibility to visit different GPs? What are insurance restrictions, if any, on seeking providers? You provide some insight into the health system now in the strengths and limitations section, but I recommend providing some details earlier in the paper.

Methods

1. Clarify if the GP was present while the intern collected data on RfE and/or symptoms and complaints.

2. Were these first visits to the particular GP? Did the survey include a question specific to the number of times the patient has seen this particular GP, and if these were follow-up visits related to the chief health complaint? If so, adding these variables to your analysis could further explain gender preferences.

3. Please explain/state whether or not incentives were given to the GPs, interns, or patients for participating in the data collection?

Discussion

1. Recommend that authors address whether the gender of the intern made a difference in patient communicated complaint or RfE, in addition to the gender of the supervising GP.

Conclusion

2. Suggest that the authors state that future studies, collecting data over several time points, could explore whether patients remained with their gender-preferred GP once the health complaint resolved.

**Do you want your identity to be public for this peer review?** For information about this choice, including consent withdrawal, please see our For information about this choice, including consent withdrawal, please see our Privacy Policy .

Reviewer #1: No

Reviewer #2: No

Reviewer #3: No

While revising your submission, please upload your figure files to the Preflight Analysis and Conversion Engine (PACE) digital diagnostic tool, https://pacev2.apexcovantage.com/ . PACE helps ensure that figures meet PLOS requirements. To use PACE, you must first register as a user. Registration is free. Then, login and navigate to the UPLOAD tab, where you will find detailed instructions on how to use the tool. If you encounter any issues or have any questions when using PACE, please email PLOS at . PACE helps ensure that figures meet PLOS requirements. To use PACE, you must first register as a user. Registration is free. Then, login and navigate to the UPLOAD tab, where you will find detailed instructions on how to use the tool. If you encounter any issues or have any questions when using PACE, please email PLOS at figures@plos.org . Please note that Supporting Information files do not need this step.. Please note that Supporting Information files do not need this step.

---

## [Author Response · Author response to Decision Letter 1]

12 Jan 2026

Dear Editor and Reviewers,

Thank you for taking the time to review our article, and for your important remarks. You will find our point by point answers to the comments made in the Decision Letter below, in the body text. I have also joined a document in the "Attach files section" with these anwers highlighted in color for more clarity.

Journal Requirements:

Thank you for these important remarks. We have added information about patients' consent to participate in the study:

[Line 113 from the Methods section/Study design] :

Patients were given oral and visual information about the study, including the presence of the intern. Verbal consent was obtained from patients or the parents of minor participants. Patients could refuse the intern's presence while still participating in the study, or refuse both. Visits for which patients refused participation were excluded.

[Line 122 from the Methods section/Study design]

A declaration was filed with the former Advisory Committee on the Processing of Information in Health Research (CCTIRS, active at the time ECOGEN was designed) and then with the National Commission for Information Technology and Civil Liberties (CNIL) – both advisory boards approved the research methodology. Further, the ECOGEN survey was approved by Ethics Committee Sud-Est IV (No.L11-149), which explicitly approved the oral form of consent obtained from patients

3. In this instance it seems there may be acceptable restrictions in place that prevent the public sharing of your minimal data. However, in line with our goal of ensuring long-term data availability to all interested researchers, PLOS’ Data Policy states that authors cannot be the sole named individuals responsible for ensuring data access (http://journals.plos.org/plosone/s/data-availability#loc-acceptable-data-sharing-methods).

Thank you for this remark. The data is available by request at the Université Claude Bernard Lyon 1 (UCBL), where they are stored on the UCBL servers.

Thank you for this remark. We have modified our statements and added the information to the article.

[Ligne 359 from Disclosure statement]

“The ECOGEN study was supported by the French National College of teachers in general practice via a grant from Pfizer laboratories (grant number not applicable), received by Laurent Letrillart. Eugenia Alcalde was funded by the Gender and Health Inequalities (GENDHI) project (grant agreement No. [ERC-2019-SyG n° 856478]). This funding had no role in study design, data collection and analysis, decision to publish, or preparation of the manuscript.”

Thank you for this remark.

a) The ECOGEN study was supported by the French National College of teachers in general practice via a grant from Pfizer laboratories (grant number not applicable), received by Laurent Letrillart.

Eugenia Alcalde was funded by the Gender and Health Inequalities (GENDHI) project (grant agreement No. [ERC-2019-SyG n° 856478]). This funding had no role in study design, data collection and analysis, decision to publish, or preparation of the manuscript.

The rest of the authors had no relevant or material financial interests that relate to the research described in this paper

c) None of the authors received a salary from the funders.

We have included the amended statement to our revised cover letter.

Reviewers' comments:

Reviewer's Responses to Questions

Comments to the Author

1. Is the manuscript technically sound, and do the data support the conclusions?

Reviewer #1: Partly

Reviewer #2: Partly

Reviewer #3: Yes

2. Has the statistical analysis been performed appropriately and rigorously?

Reviewer #1: Yes

Reviewer #2: Yes

Reviewer #3: Yes

3. Have the authors made all data underlying the findings in their manuscript fully available?

Reviewer #1: Yes

Reviewer #2: Yes

Reviewer #3: Yes

4. Is the manuscript presented in an intelligible fashion and written in standard English?

Reviewer #1: Yes

Reviewer #2: Yes

Reviewer #3: Yes

5. Review Comments to the Author

Reviewer #1: The study topic is good, but it is not completely clear and needs to be clarified, especially with regard to the study summary.

The study needs to detail the methodology, the summary, and even the results, and why gender affects medical issues.

We’d like to thank the reviewer for this global comment. We agree that our summary lacks clarity. We have modified all sections as follows:

Background: Evidence has shown that the gender of general practitioners (GPs) and of patients plays a role in primary care, both in communication and practice style. However, less is known about how both genders interact in the reasons for encounter (RfE). Studying RfEs by both patient and GP gender may help elucidate what symptoms or complaints lead women and men to consult a GP and specifically, if patients have preferences regarding the gender of their GP for specific problems. Given that men tend to consult less frequently and are often diagnosed at later stages of disease, it is essential to disentangle whether GP gender may act as a barrier for men in disclosing certain health concerns.

Then, in the methods section:

Methods: This is an ancillary study from the French ECOGEN survey, an observational cross-sectional multicentre study conducted between November 2011 and April 2012. 54 GP interns working alongside 128 GP supervisors collected RfEs during consultations. Using mixed models, we first assessed the association between patient gender and each RfE chapter (i.e., 13 independent variables) among all patients. We then stratified models according to GP gender. Finally, we compared the four physician–patient gender dyads (male physician–male patient, male physician–female patient, female physician–male patient, and female physician–female patient) in pairwise analyses.

The results section:

Results: Overall, 20 613 patients were included, of whom 58% were women. Of the 128 participating GPs, 85 were men. Women patients had higher presentation rates across all chapters. The most common RfE were those related to the respiratory system. Modest differences in presentation rates were observed between men and women when visiting male GPs, with men being less likely to present RfEs from the General and Unspecified, Digestive, Eyes, Ears, Neurological, and Respiratory chapters.

And finally the conclusion statement:

Conclusions: Our study adds evidence on how gender plays a role in presenting or disclosing health concerns in primary care. Specifically, our findings suggest that male patients may be reluctant to present certain RfEs to male GPs. Acknowledging these gender effects in primary care may help strengthen the partnership between GPs and patients.

We have also revised certain paragraphs in the introduction to clarify the topic and the relationship between gender and medical issues.

[Line 76, Introduction]:

‘One interesting way to analyse this gender gap is to investigate reasons for encounter (RfE), particularly symptoms and complaints, as they reflect what prompts men and women to seek care and what they decide to disclose to their GPs. In other words, RfEs provide insight into how individuals evaluate and prioritize health concerns in their decision-making process to seek help’.

[Line 102, Introduction]:

‘Previous studies investigating the effect of patients’ and GPs’ gender on RfE usually present results based on one or the other, but rarely together. As both seem to influence the care process, the challenge lies in distinguishing how they operate within a wide range of RfEs presented in primary care. Given that men consult less frequently and are often diagnosed at later stages of disease, it is important to disentangle whether GP gender may act as a barrier for men in disclosing certain health problems’.

Reviewer #2: The article provides a relevant analysis of gender differences in reasons for encounter in primary care, based on a large and representative dataset from France. The findings are valuable and the manuscript is well structured.

However, it is important for the authors to clarify that the results are not directly generalizable to the present context, since gender ideologies and the distribution of male and female physicians have changed in the last decade. In addition, the manuscript would benefit from a deeper consideration of the interaction between gender and age, for example discussing whether younger men and older men differ in consultation patterns, or explaining why this aspect was not explored.

Overall, the study is suitable for publication and represents a relevant contribution, but it would be strengthened by these additions to the discussion.

We’d like to thank the reviewer for this valuable comment. We agree with the fact that our study is likely not generalizable to the present context and that we lacked a more engaged discussion about age and gender interaction. We added these elements in the limitation section:

“Regarding our analyses, for simplicity reasons, our models were adjusted for patients’ age, but this did not allow us to assess the interaction between patients’ age and gender when presenting a given RfE. Indeed, it is possible that younger men or women have different attitudes towards GPs when presenting their health problems compared to older counterparts. We believe further studies should address this gap, especially considering how cohort effects regarding gender attitudes have changed in recent years.”

About the generalizability to our current context:

“Finally, the analysed data were collected over 10 years prior to our study. Since then, many changes in society and medical demographics have occurred, including a potential increase in gender awareness among patients and GPs, as well as the continued feminization of the GP population (31), which makes our results less generalizable to the current context. At the same time, the number of GPs in France declined in recent years until 2021 (32). Although more recent data suggest a slight increase since then (31), the reduced density of GPs per capita (33) may lead people to consult a GP based on availability. For these reasons, the gender of GPs may have less impact on RfEs than before, as people have fewer options. Still, using data from the

---

## [Decision Letter · Decision Letter 1]

10 Mar 2026

Reasons for encounter in primary care among French patients: gender differences in presentation rates more pronounced among the patients of male practitioners

PONE-D-25-38524R1

Dear Dr. Patrice,

We’re pleased to inform you that your manuscript has been judged scientifically suitable for publication and will be formally accepted for publication once it meets all outstanding technical requirements.

Kind regards,

PLOS One

Additional Editor Comments (optional):

Reviewers' comments:

Reviewer's Responses to Questions

**Comments to the Author**

Reviewer #3: All comments have been addressed

2. Is the manuscript technically sound, and do the data support the conclusions?

Reviewer #3: Yes

3. Has the statistical analysis been performed appropriately and rigorously?

Reviewer #3: Yes

4. Have the authors made all data underlying the findings in their manuscript fully available?

Reviewer #3: Yes

5. Is the manuscript presented in an intelligible fashion and written in standard English?

Reviewer #3: Yes

Reviewer #3: I have no additional comments. I am pleased that the authors carefully considered my previous comments and appropriately addressed them.

**Do you want your identity to be public for this peer review?** For information about this choice, including consent withdrawal, please see our For information about this choice, including consent withdrawal, please see our Privacy Policy .

Reviewer #3: No

---

## [Editor Report · Acceptance letter]

PONE-D-25-38524R1

PLOS One

Dear Dr. Patrice,

I'm pleased to inform you that your manuscript has been deemed suitable for publication in PLOS One. Congratulations! Your manuscript is now being handed over to our production team.

Kind regards,

on behalf of

Dr. Li-Da Wu

Academic Editor

PLOS One